# Bacterial cellulose spheroids as building blocks for 3D and patterned living materials and for regeneration

Joaquin Caro-Astorga [1,2], Kenneth T. Walker[1,2], Natalia Herrera[3], Koon-Yang Lee[3] & Tom Ellis [1,2 ✉]

Engineered living materials (ELMs) based on bacterial cellulose (BC) offer a promising avenue for cheap-to-produce materials that can be programmed with genetically encoded functionalities. Here we explore how ELMs can be fabricated in a modular fashion from millimetre-scale biofilm spheroids grown from shaking cultures of *Komagataeibacter rhaeticus*. Here we define a reproducible protocol to produce BC spheroids with the high yield bacterial cellulose producer *K. rhaeticus* and demonstrate for the first time their potential for their use as building blocks to grow ELMs in 3D shapes. Using genetically engineered *K. rhaeticus*, we produce functionalized BC spheroids and use these to make and grow patterned BC-based ELMs that signal within a material and can sense and report on chemical inputs. We also investigate the use of BC spheroids as a method to regenerate damaged BC materials and as a way to fuse together smaller material sections of cellulose and synthetic materials into a larger piece. This work improves our understanding of BC spheroid formation and showcases their great potential for fabricating, patterning and repairing ELMs based on the promising biomaterial of bacterial cellulose.

[1] Department of Bioengineering, Imperial College London, London, UK. [2] Imperial College Centre for Synthetic Biology, Imperial College London, London, UK. [3] Department of Aeronautics, Imperial College London, London, UK. ✉email: t.ellis@imperial.ac.uk

Engineered living materials (ELMs) are those containing cells on or within the material that play a role in its functionalisation or can produce the material itself[1–4]. Bacterial cellulose (BC) is a carbohydrate polymer produced by many bacterial species as a structural element of their biofilm and offers excellent opportunities for developing new ELMs[5]. In the past decade, progress in understanding and producing BC has now led to its use in a broad range of applications, including products used in textiles, cosmetics, healthcare, audio-visual technology and architecture[6–9]. The BC produced by several *Acetobacteriacea* species are of particular interest as these are quickly and cheaply made as pellicles—a large mass of thick BC—when the cells are grown in static rich media[10,11]. BC inherently has attractive high-performance mechanical properties and crystallinity, has a high water-retention capacity and is ultra-pure compared to plant cellulose[12,13]. These outstanding properties of BC make it an excellent candidate for developing new materials with improved technical and environmental benefits. Most of applications use sterile, purified BC as a bulk specialised material; however, BC has also shown promise as an ELM[14,15]. In one recent example, incorporating *Bacillus subtilis* cells into BC-based wound dressings helped to prevent wound infections by blocking the growth of several pathogenic bacteria[16].

Two desirable features of ELMs not routinely seen in normal materials are modular design and regeneration in response to damage. Easy and cheap repair of damaged materials (or their automatic regeneration) is an important consideration for the sustainability of all new materials[17]. BC offers excellent opportunities in this regard, because the bacteria trapped in the grown material have the potential to regenerate it by further growth and cellulose production in the future. Hypothetically, by providing nutrients, water and oxygen, the bacteria can keep growing and seal gaps and tears when they arise, so long as the material has not been dehydrated or sterilised after growth. For patterned functionalities, this can also theoretically be achieved with BC-based materials by growing these from genetically engineered cells[5]. However, another possibility to tackle this problem could be to use modular ELM building blocks and pattern these physically as a mosaic to make larger materials. Such a 'building block' approach to novel materials has been taken before in nanotechnology to increase the complexity of materials and to facilitate industrial scaling of complex pieces[18]. Modular BC-based building blocks have not been explored before in an ELM context, but BC and in particular its rapid production from living cells within the material structure offers an excellent opportunity to tackle this challenge.

Past work has shown several solutions for building BC into shapes other than the standard flat pellicles. Growing BC in hyper-hydrophobic moulds has allowed researchers to create a versatile range of three-dimensional (3D) shapes with high accuracy[19,20]. However, moulds are limited in what they can achieve in terms of size, require mould design and fabrication and typically work just for growing one material at a time. Creating patterns of functionalised BC grown from several different cell types would prove unattainable with this approach and so limits its use for creating 3D functionalised patterned BC-based ELMs. 3D printing of cells with semi-solid growth support materials is more promising in terms of creating 3D ELMs incorporating multiple strains in patterns[21]. However, the additive manufacturing approach relies on specialised equipment and the majority of the product is support material, rather than living material. A building-block approach where modular units of living material grow and self-connect into 3D shapes would be less cumbersome.

BC grows as floating pellicles in the air–liquid interphase of glucose-rich media. Under shaking conditions, some strains may produce millimetre-scale rounded BC particles named variously in the literature as spheroids, spherical granules, sphere-like BC or sphere-like BC particles[22–24]. Here we introduce the concept of using BC spheroids as a building block to make and engineer 3D BC-based ELMs. Growth of BC spheroids has remained poorly understood and is typically characterised as being strain dependent or inconsistently produced[22]. Here we now define a reproducible method to produce BC spheroids from the bacterium *Komagataeibacter rhaeticus*. We use these BC spheroids as building blocks to build 3D shapes and to create patterned ELMs made of spheroids containing genetically functionalised bacteria that impart fluorescence and send and receive signals. We further demonstrate that spheroid building blocks can be used to regenerate damaged BC materials as fresh, stored or purified spheroids and that this can be used to fuse and assemble independent BC pieces in a mosaic fashion.

## Results

**BC spheroids**. BC spheroids have been reported in several previous studies[22–24], but how they form and why has not been fully elucidated. It has been hypothesised that spheroid formation is produced by the adhesion of bacteria to air bubbles produced in shaking media, with cellulose then grown at the bubble air–liquid interphase to form a spheroid shape[25]. Past experiments in our laboratory growing *K. rhaeticus* in shaking conditions occasionally produced spheroids. As with other BC-producing bacteria, we originally assumed that spheroid formation by *K. rhaeticus* occurred randomly, either in response to stochastic processes during shaking growth or due to mutation or another form of uncontrolled variation in cell behaviour that triggers their spontaneous production.

Here we set out to examine whether spheroid production by *K. rhaeticus* was indeed a random event or one that could be reproducibly triggered. To do this, we grew *K. rhaeticus* cells with shaking at 30 °C and tried combinations of >20 different growth variables (Supplementary Table 1). During and after growth, we visually assessed the cultures for the presence of BC spheroids and used this information to determine the key factors involved in spheroid formation and which combination of growth variables leads to reproducible growth of BC spheroids.

Our results indicated that there are two critical factors required simultaneously for BC spheroid formation. The first is the initial optical density (OD) of the culture, with more ideal spheroids seen when cultures begin at low optical densities ($OD_{600} = 0.001–0.0001$) where bacteria are more likely to begin isolated from one another (Supplementary Fig. 1A). The second most important factor seen in our experiments was the culture container. BC spheroids were only commonly observed after shaking growth in 1.5 cm diameter 14 ml round base and 1.5 cm diameter 15 ml conical base plastic culture tubes. On some occasions, spheroids appeared in 25 ml tubes (2 cm diameter) but never grew in any attempts with 50 ml tubes or with larger bottles and flasks. Oxygen diffusion was considered as a possible explanation, but shallow cultures using low media volume were also unsuccessful, leading us to propose the hydrodynamic effect and media motion as the critical factors. The third factor revealed in our experiments affecting spheroids formation was the culture media. The use of 2× Hestrin–Scharmm (HS) instead of normal HS media produced a higher yield in the number of spheroids per tube, but the spheroids were smaller in size and less uniform (Supplementary Fig. 1B). Addition of 1% ethanol has been reported to increase the BC yield when growing static pellicles[26], but we found in multiple trials that ethanol usually inhibits spheroid formation.

With these three factors determined, it became possible for us to produce a protocol for reliable spheroids production (Fig. 1a) in

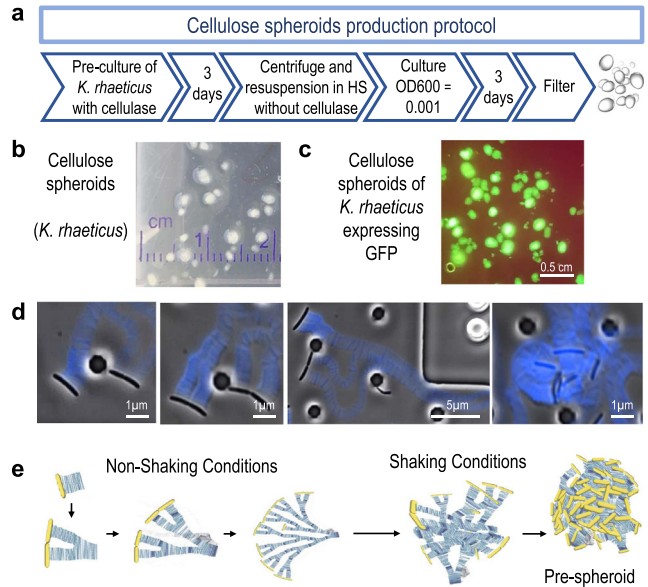

**Fig. 1 Bacterial cellulose (BC) spheroid production. a** Schematic of the protocol steps to produce BC spheroids from *K. rhaeticus* in shaking cultures. **b** Example image of BC spheroids produced by wild-type *K. rhaeticus* cells. **c** Example fluorescence image of BC spheroids produced by superfolder GFP-expressing *K. rhaeticus* cells. **d** Microscopic images of a microfluidic growth time-lapse of *K. rhaeticus* cells growing at low density with calcofluor blue staining of cellulose bands. Images show the first stages of BC spheroid development. Images are representative of four experiments. **e** Schematic showing the growth progression of cellulose in static and shaking conditions, hypothesised to lead BC spheroid formation.

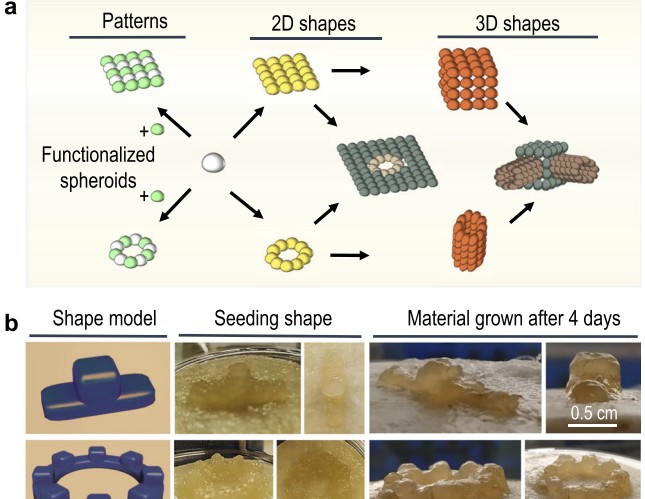

**Fig. 2 BC spheroids as 3D building blocks. a** Schematic of potential 2D and 3D structures that can be built from spherical building blocks. **b** Growth of two example 3D shapes constructed using BC spheroids (see Supplementary Materials for further examples). Model of desired structure (left), the seeded spheroid shape assembled manually (middle) and the resultant material structure produced after 4 days of further growth (right).

14 ml culture tubes, yielding spheroids typically 0.2–2 mm in diameter after 3 days of culture (Fig. 1b). We also tested whether our protocol for spheroid cultivation would also work for genetically modified strains of *K. rhaeticus* containing plasmids expressing transgenes. We also observed reliable growth of fluorescent spheroids from engineered *K. rhaeticus* expressing superfolder green fluorescent protein (sfGFP) and monomeric red fluorescent protein (mRFP) genes from plasmid constructs (Fig. 1c).

In order to observe how BC is produced by our bacteria, we performed microfluidic time-lapse culture experiments with calcofluor added to stain nascent cellulose production (Supplementary Video 1). We observed band-like growth of BC chains coming from one side of the bacteria longitudinal axis, producing branches of cellulose as the cells divide (Fig. 1d). In static culture, this event is responsible for the formation of layers of cellulose growing into pellicles. However, when perturbed by shaking, the branches collapse on themselves, entangling the BC bands while the chains continue growing and cells dividing. Although conditions in the microfluidic chamber do not match those used for spheroid production in our protocol, we reason that the same processes lead to the formation of spheroids. When cultures are seeded at very low density, BC bands interact with themselves, producing a 'pre-spheroid' mass that develops into a spheroid. We speculate that higher cell densities favour interactions between the cellulose of different pre-spheroids, entangling them at an early stage and yielding amorphous fibrous clumps, rather than spheroids (Fig. 1e).

**Construction in two-dimensional (2D) and 3D with BC spheroids**. Given the mechanism of growth, we reasoned that spheroids would continue cellulose production at their surfaces and thus when two spheroids interact for enough time they will grow together and fuse. This property of our spheroids would allow them to act as millimetre-scale BC-based building blocks that could then be used to produce 2D and 3D shapes of desired designs (Fig. 2a). To demonstrate this application, we designed three 3D shapes of different complexity, a podium, a serrated ring and a pentagon with elevated apexes. We then manually placed spheroids in these arrangements on sterile cotton or paper pads, with the help of a sterile pipette tip (Supplementary Fig. 2). After 4–10 days of incubation at 30 °C, the spheroids had visibly grown and fused together to create continuous BC-based shapes roughly matching the seeded design (Fig. 2b). Although material growth from the spheroids within the structure is reduced due to the reduced oxygen diffusion, the BC shapes that grow are flexible, interconnected and resistant to deforming (Supplementary Video 2).

The use of cellulose spheroids as building blocks opens a new opportunity to create ELMs with 2D and 3D patterns with functional properties genetically encoded into the bacteria within the material. As *K. rhaeticus* is a non-motile bacteria, the patterns created from spheroid seeding will not get blurred at the macroscale and should remain roughly conserved. To demonstrate this, we designed and created layers of BC seeded onto filter paper with both normal BC spheroids and BC spheroids made by GFP-tagged *K. rhaeticus*. Although BC has some green autofluorescence, it is very low compared to GFP signal and can be removed by setting the signal threshold. We placed the fluorescent spheroids in three lines close to each other, setting a pattern to make diagonal lines of non-fluorescent spheroids within the pattern. After 10 days of incubation at 30 °C, we obtained a fused pellicle conserving the fluorescent pattern of seeding, demonstrating the possibility of easily creating ELMs functionalised at the millimetre scale—the diameter of a single spheroid (Fig. 3a).

We observed in these experiments that growth after seeding also led to the spheroid structure being adhered to the sterile filter paper support. As paper is itself predominantly cellulose, we wondered how the spheroids would behave if they were set to grow on a layer of BC. To examine this, we seeded fluorescent spheroids on a normal *K. rhaeticus* pellicle produced from static growth. After 4 days of incubation at 30 °C, the spheroids were completely fused

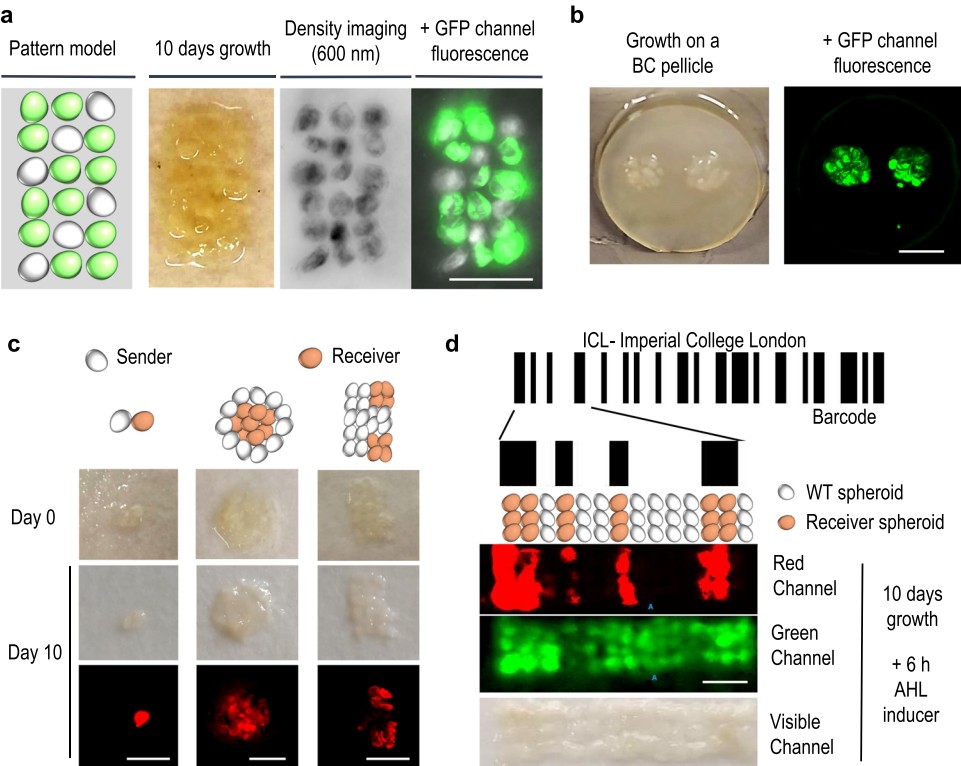

**Fig. 3 Patterned materials with functionalised bacterial cellulose (BC) spheroids. a** Patterns of functionalised BC spheroids produced by GFP-expressing bacteria. Intended pattern model (left), visible channel image of pellicle after 10 days growth (middle left), spheres imaged using 600 nm excitation laser (middle right) and the merged 600 nm image and green fluorescence imaging channel (far right). **b** Mosaic of GFP-expressing BC spheroids grown for 4 days at 30 °C on an existing bacterial cellulose pellicle. Images of the material after 10 days of growth in the visible channel (left) and green fluorescence channel (right). **c** Patterns built with two different genetically functionalised spheroids that secrete (sender) and respond (receiver) to acyl-homoserine-lactone (AHL). Receiver spheroids express RFP in response to AHL produced by neighbouring sender spheroids. Image of initial patterned material (top), visible channel image after 10 days (middle) and red green fluorescence channel after 10 days (bottom). **d** Example of a functionalised BC spheroid pattern; a chemically responsive barcode. The Code-128 barcode for ICL (Imperial College London) was generated and a section of this was recreated as a BC pattern using a grid of 14 × 3 spheroids grown from either wild-type bacteria or the receiver bacteria used in **c**. Spheroids were grown for 10 days into a continuous material appearing white/brown to the naked eye. Soaking this in 25 nm AHL for 6 h triggered RFP expression from the receiver bacteria in the pattern. The hidden barcode pattern is then revealed by scanning for red fluorescence, whereas scanning for green fluorescence only reveals the spheroid density. Scale bars are 5 mm.

to the base pellicle, revealing that a BC layer provides an excellent frame within which to set spheroids in patterns. Fluorescence was localised only in the area of seeding (Fig. 3b).

We next wondered whether spheroids containing engineered cells could work as functional entities able to interact and respond to one another. To demonstrate this, we assembled patterns using two different functionalised spheroids, one grown from bacteria engineered to produce and secrete acyl-homoserine-lactone (AHL) (pSender spheroids) and one grown from bacteria engineered to sense AHL levels and in response express mRFP (pReceiver spheroids). These bacteria were constructed and characterised by our laboratory in a previous study[5]. Using just one of each spheroid placed adjacent in the material was enough for the AHL-secreting pSender spheroid to activate its neighbour pReceiver spheroid. Seeding in a circular distribution surrounding a group of receiver spheroids was further used to show how the signal diffuses to spheroids even when they are not in close contact. Seeding in a squared distribution allows to us distinguish the distribution of sender and receiver spheroids (Fig. 3c). More complex patterns were achieved with more spheroids, and the complexity of these patterns could theoretically be extended further by the addition of spheroids grown from other engineered

bacteria, such as those expressing relay circuits that sense AHL and produce more AHL in response to extend the range of the signalling[27].

As a demonstration of how spheroids can enable smart, reactive materials, we prototyped a method whereby a grown material can reveal hidden information upon chemical input. We designed a simple barcode and used this to define a 2D pattern that we then fabricated from BC spheroids and grown into a continuous material. Spheroids grown from wild-type bacteria were used to seed white space in the pattern, and spheroids grown from pReceiver bacteria were used to see black space. After 10 days of growth, the material was uniform and uncoloured, with no pattern visible to the naked eye. However, after 6 h exposure to 25 nM AHL, the pReceiver bacteria within the material produced RFP, which then revealed the barcode pattern when imaging in the red wavelength emission channel (Fig. 3d). Autofluorescence of cellulose in the green channel not only verifies cellulose density in the grown material but also illustrates that visualising the pattern of the material requires the correct emission wavelength. Only with prior knowledge of how to chemically induce the expression and how to detect it can the user access the hidden barcode information.

**Material repair and fusion using BC spheroids**. Given the interest in using BC as a basis for ELMs, there is a need to identify methods to repair or regenerate a BC-based material when damaged. To investigate this, we established a repair assay of BC pellicles using a mechanical hole punch to damage the material. We first assessed whether just the addition of HS media and further incubation for 7 days in static aerated conditions would result in regrowth of BC in the hole wound of a punctured fresh BC pellicle. Although occasionally a thin BC layer did grow over the hole, the new thin pellicle layer was poorly adhered to the original one below it (Supplementary Fig. 3A). We considered that the poor adherence was a result of adding too much liquid growth media, inducing the formation of a new BC layer only at the air–liquid interface and not well adhered to the original pellicle. Thus, we decided to just add a few drops of new HS media in the holes with and without also adding the circles of cellulose produced by the hole puncture (Supplementary Fig. 3B, tear and hole controls). After 7 days of incubation, this still failed to show stable wound repair. To help the regeneration process, we placed growing cellulose-producing bacteria or fragments of fresh BC in different forms into the wound holes and incubated for a week (Supplementary Fig. 3B). Repair quality was assessed by a stability test, holding the pellicle edges with tweezers and pulling. Although bacteria seeded within an agar matrix and cellulose patches passed the test, there was no regrowth of material by these methods and the puncture reappeared when the material was manipulated.

The incubation of two BC pellicles stacked upon each other does not produce regrowth or their fusion but efficient fusing of spheroids into a pellicle was seen in our patterned materials (Fig. 3b). This is because BC pellicles grow anisotropically, producing new cellulose predominantly in the horizontal axis at the air–water interface and building up the pellicle by stacking new cellulose layers one over the other. In contrast, spheroids grow BC isotropically, producing it in every radial direction (Fig. 4a). This way of growth could provide a method for pellicle damage repair, fusing not only to other spheroids but also to any piece of hydrated cellulose in close contact (Fig. 4b). To demonstrate this, we placed freshly grown spheroids into puncture holes at high density and incubated for 3 days to allow for some cellulose production. We saw excellent repair that was not only stable but also restored the consistency and appearance of the material (Fig. 4c). Notably, the upper BC layers repaired with spheroids looked very similar to an unpunctured pellicle once lifted (Fig. 4d). This suggests that there is no change in the diffraction angle of the light between the original BC and newly synthesised cellulose from the spheroids, likely due to the spheroids producing similar cellulose that infiltrates surrounding materials and re-networks with its BC.

We next took these repaired pellicles and assessed their strength compared to pellicles with no puncture. Pellicles were sterilised and dried out before samples were subjected to tensile mechanical testing, cutting the testing strips within the hole and discontinuing the original BC to measure only the force of spheroids adhesion (Fig. 4e). As expected, the repaired material was weaker, but impressively it still retained 40% of its tensile strength and its Young's modulus was 58% of that for the undamaged material.

Given that strong repair of punctured material was possible, we reasoned that BC spheroids could also be used to fuse sections of BC material together. As a proof of concept, we grew four small square pellicles and then placed them adjacent to each other and filled the space in between with fresh BC spheroids. After 10 days of incubation on each side, the four squares were now fused into a 'mosaic' that was strong enough to hold its own weight when handled with forceps (Fig. 4f). This same property was then demonstrated with several other materials. After only 5 days of incubation, sterilised pieces of sponge, wood and cotton were completely fused to each other by growth of BC from spheroids placed between sections (Fig. 4g, Supplementary Fig. 5A and Supplementary Video 3). The tensile strength of the fusion was measured, obtaining different values depending on the material (0.1–5 MPa) and revealing low tensile strength values compared with the outstanding properties of BC material (Supplementary Fig. 5B). The fused material always broke at the joint, which means that the integration method could be improved, for instance, in the edge shape of the pieces to increase surface contact. Spheroids therefore offer the opportunity to integrate BC ELMs into other porous natural or synthetic materials, increasing the possibilities of combining and mixing material properties and genetic engineering.

**Regeneration from stored BC spheroids**. To better understand the regeneration process, we explored the colonisation capacity of *K. rhaeticus* in repaired pellicle pieces. To enable visualisation of the process, we used spheroids made from GFP-expressing bacteria to repair a sterile, purified pellicle of BC. Visualisation of the green fluorescence revealed that the bacteria spread into the surrounding areas (Fig. 4h, image settings thresholded to reduce cellulose autofluorescence). However, when repeated with a fresh pellicle still with an active bacterial population, colonisation was constrained only to the local area (Fig. 4i). The pellicle was still repaired as expected, but the bacteria did not infiltrate as far, possibly due to competition for space or another method of exclusion.

The importance of finding a methodology to repair BC materials led us to investigate *K. rhaeticus* survival in BC and whether BC spheroids used for repair could be made and stored in bulk in advance. Assessing cell survival within the spheroids proved difficult due to the small size of spheroids and irregular shapes giving variable volumes and thus cell counts. Therefore, we instead used small BC pellicles grown in 96-well microplates as an equivalent material. After growth of this BC material, the small pellicle samples were immediately stored in either vacuum-sealed bags (stored at 4 and 23 °C) or stored in 2 ml tubes at 23 °C. Then, over a period of time, the samples were removed from storage, digested with cellulase and grown on solid agar plates in order to determine cell number per sample by calculating colony-forming units (CFUs). This revealed that the number of viable cells in the material decreased rapidly in the first 3 weeks for all three storage methods tested (Supplementary Fig. 4). However, viable cells were still recovered beyond 12 weeks in all cases, and when the BC was vacuum-sealed and stored at 4 °C, cells were still recoverable beyond 6 months.

Spheroids grown from GFP- and RFP-expressing bacteria were stored for 3 months at 4 °C in culture tubes and then used to repair a puncture in a fresh BC pellicle. Remarkably, this produced the expected pellicle repair. However, fluorescence was almost completely lost (Fig. 4i and Supplementary Fig. 6). This may be explained by loss of the plasmid harbouring the fluorescent genes from the bacteria in the spheroids or from the death of most of the bacterial cells. Considering that the fluorescence was localised just to one or two spots, we reasoned that cell death was the most likely explanation. The fact that repair was successful, even with most dead spheroids, suggests that in this case bacteria from the fresh pellicle are migrating to the spheroids and producing the new cellulose. To verify this, we purified sterile BC spheroids and seeded these in punctured fresh pellicles produced by GFP- and RFP-expressing bacteria. As expected, the pellicles were repaired, and fluorescence was also observed in the repaired holes, in line with the hypothesis that the repair is performed by the pellicle bacteria in this set-up. (Fig. 4i and

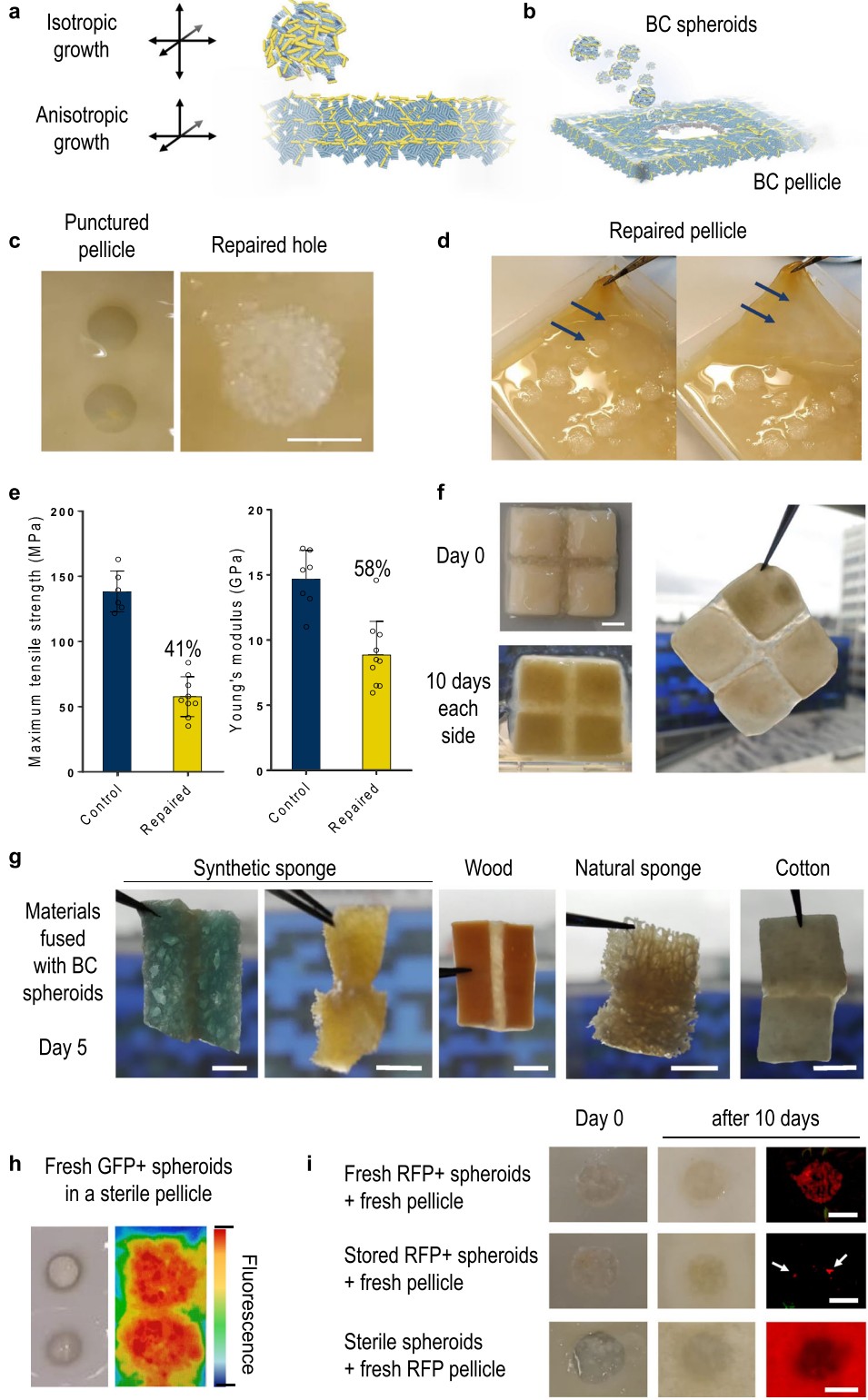

6        NATURE COMMUNICATIONS | (2021)12:5027 | https://doi.org/10.1038/s41467-021-25350-8 | www.nature.com/naturecommunications

Supplementary Fig. 6A–C). Although BC-producing bacteria are not actively motile, they can migrate at the microscale into surrounding materials due to propulsion from their cellulose production[28]. However, a further explanation is that the sterile spheroids may get inoculated with new bacteria from the culture media of the fresh pellicles. To assess this, we repaired a purified sterile pellicle with purified sterile BC spheroids, this time just by adding 50 µl of live

bacteria suspension ($OD_{600} = 1$; Supplementary Fig. D and Supplementary Video 4). In terms of repair, this means that BC spheroids can be stored sterile and bacteria stored separately (e.g. frozen in 15% glycerol) and the combination together can be used for material repair.

However, in some instance, it would be desirable to store the living bacteria within the material for future use. We therefore

**Fig. 4 Repair of BC materials using bacterial cellulose (BC) spheroids. a** Illustration of the BC growth axis of spheroids compared to those seen in flat pellicles. **b** Schematic of the regeneration strategy of repair by seeding punctured holes with spheroids. **c** Image of punctured holes in BC pellicles (left) and result of the repair of using dense packing of BC spheroids (right). **d** State of a repaired BC pellicle showing site of puncture repairs (left, arrows) and the lack of the holes when the layer is elevated (right). **e** Results of the mechanical tests performed on repaired BC pellicle samples in comparison to controls. Tensile strength (left) and Young's modulus (right) are shown, error bars indicate SEM of 6 controls and 9 samples in 3 different pellicles. Source data are provided as a Source data file. **f** Result of the fusion of four 0.1 mm thick BC sections using BC spheroids. Sections were seeded with spheroids in the gaps at Day 0 (top left) and incubated to grow for 10 days on one side and then 10 days on the other side to achieve a fused mosaic (bottom left). The final material is shown held from forceps (right). **g** Images of materials fused from BC spheroids growing between pieces for 5 days at 30 °C. **h** Bacterial dynamics of BC repair. Images of two purified hole-punched BC pellicles repaired using GFP-expressing bacteria (left: visible channel, right: fluorescence channel) showing the degree of bacterial colonisation. Colouring indicates intensity of fluorescence (highest in red, lowest in blue), revealing the greatest intensity inside the seeded spheroids and measurable signal corresponding to the colonisation of the local region of pellicle. **i** Regeneration at BC pellicle punctures using fresh (top), stored (middle) and sterile (bottom) BC spheroids. Left images show visible channel at the day of seeding, middle images show visible channel after 10 days incubation, right images show the fluorescence channel after 10 days, imaging for red fluorescent protein. Fresh and stored spheroids were made by RFP-expressing cells, with stored spheroids used after 3 months of storage at 4 °C. The sterile spheroids were made from non-engineered cells and purified before use to repair a fresh pellicle grown from RFP-expressing cells. Scale bars are 5 mm.

explored the possibility of using glycerol as a preservative agent to freeze live pellicles. Freshly grown BC material was soaked in 15% glycerol and then stored at −80 °C. To assess survival, we counted bacteria from digested material samples from before glycerol treatment and after 1 h in glycerol 15% solution at 20 °C and then after freeze storing for 1 h, 1 week and 1 month. No significant different was found between cell counts, confirming that the method is effective for bacteria conservation within BC. This finding enables us to consider long-term conservation of BC-based ELMs (Supplementary Fig. 7).

## Discussion

Previous research on BC spheroids has stated that their production is strain dependent with some strains capable of their production but others not. For example, the *Gluconacetobacter xylinus* JCM 9730 strain was shown to produce spheroids but *G. xylinus* NCIMB strain did not[22]. In our experiments, when the shaking culture starts at high density, we always fail to obtain spheroids and instead obtain amorphous aggregates. This tallies with previous work with *G. xylinus* JCM 9730 which showed that differences in the number and size of spheroids grown depended on the volume used to inoculate the culture[29]. While previous work has hypothesised that attachment and growth around air bubbles triggers spheroid production[25], our data offer a new insight, showing an entangling process of cells and cellulose chains during the early stages of culture growth. When cell densities are high, we expect that this causes entangling of mass before individual spheroids can form. The volume and shape of the container also affected spheroid formation, likely due to how the motion of the growth media is affected. We showed that having a consistent low number of cells to begin the culture is critical. We note that previously explored methods for spheroid formation have not measured the cell density of the bacteria inoculum before beginning culturing. It may be the case that all BC-producing bacteria can produce BC spheroids if seeded at the correct low density, and we speculate that other characteristics like shape, size or density of cellulose could also be customised with more research into growing conditions.

Our work also demonstrated the potential for BC spheroids to be used as building blocks to create 2D and 3D shapes and create patterns that could find use as functionalised ELMs. The building block approach opens a myriad of applications, especially when considering the possibility of using BC spheroids grown from genetically reprogrammed cells that perform desired tasks. By taking a synthetic biology approach, we here engineered *K. rhaeticus* to produce, sense and respond to AHL, but in the future, these cells could be reprogrammed with a plethora of other modular DNA parts. Many have been described in previous

work that enable bacteria to sense a wide variety of biological, chemical and physical inputs and to signal to one another in ways analogous to electric circuitry[30–33]. Different functionalised spheroids engineered with such modular DNA parts can be used to fabricate 2D and 3D patterned materials that compute environmental information and in response reveal barcodes, QR codes or symbols displaying useful visible information or remain hidden and can be read with the appropriate excitation laser and receptor. This property could have direct application to sense infections or progress of the tissue repair in wound-healing BC applications. Or even for BC patches for skin in which multiple functionalisation with spheroids could sense sweat biomarkers, a new developing field able to report signals related to infections, immunology, neurology, endocrinology, nephrology, oncology and pharmacology[34]. With our method for bacteria and BC conservation in using glycerol storage, we provide a useful method for conservation of ELMs without consuming nutrients, overcoming a key limitation of these materials for storage and end-use.

Patterned materials grown from engineered cells also offer great promise as a future biomedical technology. Spheroids grown from engineered cells designed to also secrete proteins could be used to grow 2D and 3D materials where different regions of the material display anchor proteins or growth factors that attract mammalian cells and promote their differentiation. Such a material could, for example, be used to seed and grow complex mammalian tissues like skin or cartilage in defined patterns and layers and contribute to organoid production.

However, for creating small 3D shapes, BC spheroids are limited as building blocks by their millimetre size and the low precision of fabrication by hand. For making smaller or more precise BC-based ELM shapes, it may be more convenient to use moulds or recent 3D printing methods developed for bacterial cultures, such as the FLINK method where a non-living gel matrix that harbours the bacteria is printed into the desired shape and the cells then grow and produce BC within this[21]. 3D printing also allows the production of functionalised ELMs using different bacteria, but its accuracy is limited by the width of the extrusion printing head, which itself is limited by the high viscosity of the printed gel. A future solution to this problem could be a hybrid approach where a 3D printer is programmed to precisely dispense BC spheroids with different functionalisation (e.g. sensing, binding, reporting), embedding these in different matrix compositions. This would produce multidimensional BC-based materials and these printed sections could themselves be further fused together using BC spheroids. At what length scales (small or large) that this can be achieved remains to be explored in future work.

## Methods

**Strains, culture conditions and BC pellicle growth.** The *Acetobacteriacea* strain used in this work was *K. rhaeticus* iGEM[35]. The wild-type strain was tagged with green and red fluorescence by transformation with plasmids KTK_124 and KTK_182, respectively. These plasmids express either sfGFP or mRFP from the strong J23104 constitutive promoter (see Supplementary Fig. 8). They were constructed by Golden Gate assembly into a plasmid vector based on pSEVA331Bb (*Escherichia coli–K. rhaeticus* expression vector, ori-pBRR1 origin of replication, chloramphenicol resistance)[35]. *K. rhaeticus* cells were transformed by electroporation using 100 μl of electrocompetent cells and 20–500 μg of plasmid DNA in a 1 mm path electrocuvettes, using electroporation parameters 2.5 kV, 5–8 ms, 400 Ohm resistance and 25 μF capacitance, as described in Florea et al.[35].

Pre-cultures of *K. rhaeticus* were prepared by taking cells from −80 °C stocks and growing in 50 ml tubes with 10 ml of HS media (peptone 5 g/l, yeast extract 5 g/l, 2.7 g/l Na$_2$HPO$_4$, 1.5 g/l citric acid, 2% glucose, 2% cellulase from *Trichoderma reesei* [Sigma-Aldrich]) in shaking conditions at 250 rpm/min and at 30 °C for 3 days. 2 × HS media was used for spheroid production tests and was prepared by doubling the concentrations of peptone and yeast extract.

To grow pellicles, pre-cultures were centrifuged at 7000 × g for 3 min, and cells were then resuspended in 10 ml of HS without cellulase. This process was then repeated. The suspension was diluted 1 in 100 to grow pellicles, growing in shallow containers with 200 ml of HS without cellulase, supplemented with 1% ethanol and incubated at 30 °C for several days. To grow cellulose spheroids, the suspension was diluted 1 in 1000 in 3 ml of HS without ethanol, in 15 ml tubes, in shaking conditions at 250 rpm/min and 30 °C for 3 days using shaking incubator model Thermo Fisher MaxQ600.

**BC synthesis time-lapse.** A frozen glycerol stock of *K. rhaeticus* was used to inoculate 5 ml of HS media containing 2% glucose. The culture was incubated static for 7 days at 30 °C until a pellicle formed. To prepare the microfluidic plate, 50 μl of culture from underneath the pellicle was removed and placed into the inlet well of a B04A CellASIC ONIX plate (Merck). The plate was placed onto a fluorescence microscope (Nikon Eclipse Ti inverted microscope) within a chamber heated to 30 °C, and cells were fed with a continuous flow at 1 Psi of HS media containing 2% glucose and 0.001% Fluorescence Brighter 28 (Sigma-Aldrich). After 24 h, growing cells were identified and a time lapse was started using the bright field and DAPI channels, and images were taken every 2 min for 2 h.

**3D structures and fluorescence.** Spheroids from day 3 cultures of both wild-type and sfGFP-tagged strains were collected by filtering the culture with filter paper in sterile conditions. The spheroids were seeded in the desired shape with the help of a pipette tip and then incubated for 4 days at 30 °C. To create patterns, spheroids were taken one by one using sterile pipette tips and placed together in the desired positions.

Pellicles repaired with fluorescent spheroids and spheroid patterns made with fluorescently tagged cells were imaged with an Amersham Typhoon Scanner, using 10 μm resolution. A far-blue light gel transilluminator with amber filter was used to image spheroids produced by sfGFP-tagged cells and mRFP-tagged cells.

**Cellulose purification.** BC pellicles or spheroids were purified by soaking with 200 ml of 0.1 M NaOH solution overnight at room temperature, for 3 nights in a row in order to completely remove culture medium residues and cells. NaOH was removed by washing with distilled water overnight for 5 nights in a row.

**Pellicle repair and pellicle fusion.** For pellicle repair, a 0.8 cm hole puncture tool (Jenley hollow leather punch) was used to puncture holes in BC pellicles. Spheroids from day 3 cultures were filtered with filter paper in sterile conditions to separate them from the liquid culture. The spheroids were placed in the puncture holes and 50 μl of HS supplemented with 2% glucose was dropped over the spheroids in the holes. Two millilitres of HS was then added into the surrounding container of the pellicles to maintain them in a hydrated state. The pellicles were incubated in static conditions for 4–10 days at 30 °C before imaging.

In unsuccessful attempts of pellicle repair, the following were used (i) fragments of biofilm found adhered to the wall of a flask after 4 days in shaking conditions; (ii) floating clumps formed in shaking conditions from an initial culture set with high cell density (OD$_{600}$ ~0.5); (iii) cellulose aggregates present in the culture medium under the pellicle; (iv) a pellet of cells grown in shaking conditions with cellulase, centrifuged, washed with HS and centrifuged again; (v) cells from (iv) embedded in a 0.3% HS agar matrix at 40 °C and immediately placed in the pellicle before it solidifies; (vi) a cellulose patch of slightly bigger dimensions than the hole produced in the pellicle, used to force the edges of the patch and the hole to be in close contact. Stability test consisted of holding the pellicle edges with forceps and pulling up while observing whether the repaired material remained in position. Further manipulation and investigation of the repaired material was done to observe whether repair was from real regrowth of cellulose.

Pellicle fusion was performed using square pellicles grown in 1.2 × 1.2 cm 24-well plates for 5–7 days. As with pellicle repair, spheroids from day 3 cultures were filtered to separate them from liquid culture. They were then placed between the square BC pieces. In all, 200 μl of HS supplemented with 2% glucose was placed between the BC pieces and 2 ml of HS was added into the surrounding container to maintain a hydrated state. The pellicles were incubated in static conditions for 4–10 days at 30 °C before imaging.

Synthetic and natural sponges, cotton and wood pieces were sterilised with commercial bleach for 2 min, rinsed with water and soaked with HS media supplemented with 2% glucose. Fresh BC spheroids were placed in the intersection of the two pieces. The pieces were incubated for 5 days at 30 °C before imaging and video recording of the fused materials.

**Microtensile testing of BC pellicles.** Pellicles were dried by placing them between two stainless steel blocks and incubating the stack at 60 degrees for 48 h. Samples were taken from within the repaired holes and in between them to reduce natural variability in BC pellicles. Tensile testing of the dried undamaged pellicle and the repaired pellicles samples was conducted on miniaturised rectangular test specimens of 5 mm in width and 40 mm in length, cut using a Zwick/Roell ZCP 020 manual cutting press (Zwick testing machines Ltd., UK). Prior to the test, two dots were marked on the surface of the test specimen in the direction of the applied load. The miniaturised rectangular test specimen was then mounted onto a micro tensile tester (model MT-200 Deben UK Ltd., Woolpit, UK) equipped with a 200 N load cell. The exposed length of the rectangular test specimens was 25 mm. A non-contact video extensometer (iMetrum Ltd., Bristol, UK) was used to record the strain of the test specimen during tensile loading by monitoring the movement of the two previously marked dots. The tensile test was carried out using a crosshead displacement speed of 0.5 mm/min and a gauge length of 10 mm. Average results of five controls and six repaired samples from three different pellicles were reported. Error bars indicate SEM. All tests were performed at room temperature and relative humidity of 40%.

**Cell survival in pellicles.** A homogeneous suspension to grow pellicles was prepared as described above. Pellicles were grown in 96-squared deep-well plates at 30 °C. After 7 days, pellicles were stored in vacuum-sealed plastic bags at 4 and 23 °C and also in 2 ml tubes at 23 °C. Samples in triplicate were collected at each time point to assess survival. Pellicles were placed in 2 ml tubes, suspended in HS diluted 1 in 10 with 5% cellulase and incubated at 37 °C for 3 h in shaking conditions to degrade the cellulose. Serial dilutions of each suspension were made, and these dilutions were plated in four replicates on HS agar supplemented with 2% of glucose. After 7 days of incubation at 30 °C, colonies were counted from the agar plates and CFUs per cm$^2$ pellicle area was calculated.

**Bacteria conservation within frozen BC samples.** Pellicles were grown in 96-squared deep-well plates at 30 °C in static conditions. After 7 days, pellicles were collected and each soaked in 2 ml of 15% glycerol for 1 h, in a 2 ml tube, using an orbital shaker at 50 rpm to facilitate homogenisation. After that, pellicle samples could be stored at −80 °C. When pellicles are to be recovered from −80 °C, they must be defrosted and washed in 2 ml of HS for 1 h to dilute the glycerol.

**BC density.** Three pellicles grown in a 96-squared well plate and a similar volume of spheroids was weighed in a tared plate. The samples were dried out in an incubator at 60 degrees overnight. The difference between dry and wet weights was used to calculate the percentage w/w of cellulose/water in both samples types.

**Reporting summary.** Further information on research design is available in the Nature Research Reporting Summary linked to this article.

## Data availability

All data supporting this study are provided in full in the 'Results' section of this paper, as Supplementary Information or in the Source data file. Source data are provided with this paper.

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

## Acknowledgements

The authors wish to thank Dr. Vivianne J Goosens, Dr. Pablo Apolinar and Mr. Amritpal Singh for their advice with this project at Imperial College London. We acknowledge the US Office of Naval Research Global (ONRG) and US Army CCDC DEVCOM grant W911NF-18-1-0387 and UK Engineering and Physical Sciences Research Council (EPSRC) grant EP/N026489/1 for funding this work.

## Author contributions

J.C.-A. proposed and designed this research, performed most experiments and wrote the manuscript. K.T.W. performed microfluidic growth time-lapse experiment. N.H.V. and J.C.-A. performed tensile strength experiments and analysed data. K.-Y.L. supervised and gave advice on mechanical testing. T.E. supervised all the work and reviewed and revised the manuscript. J.C.-A., N.H.V., K.-Y.L. and T.E. discussed the results and commented on the paper.

## Competing interests
The authors declare no competing interests.
