## [Peer Review File · Nature Communications]

Reviewers' Comments:

Reviewer #1:

Remarks to the Author:

The authors present a very interesting study on the preparation and use of bacterial cellulose (BC) spheroids for patterning and regeneration of engineered living materials. This is a highly interesting topic that has a lot of potential yet to be explored. The concept proposed is to use spheroids as building blocks to make living materials with genetically programmed patterns and 3D shapes. Exploratory experiments show that a low initial cell density during culture and small culture tubes are needed to produce BC spheroids by shaking. The authors show that BC spheroids can be programmed to send and receive chemical signals, encode information that can be retrieved upon chemical activation, and repair, regenerate and glue different materials. This study is expected to motivate a lot more work in this growing research field. Before acceptance of the manuscript, the following points need to be addressed:

1. In the introduction, the authors state that additive manufacturing "requires washing the printed matrix to obtain the final shape". Since no washing is needed in the method reported in Ref 19, please indicate another reference for previous work where this step is needed.
2. The use of low bacterial densities as a criterion to form spheroids is reasonable. But why does spheroid formation depend on the volume and shape of the culture tube? Is the volume of the tube expected to change the oxygen concentration? Or is this a purely mixing/hydrodynamic effect? To allow researchers to reproduce these results, it is important that the authors provide more detailed information on the stirring conditions, e.g. shaker type, model, power, etc
3. The barcode demonstration is very nice. However, for how long would one be able to keep this functionality considering that the bacteria eventually die upon storage?
4. Figure 4H: To ensure that the fluorescent signal comes from the GFP-expressing bacteria, wouldn't it be important to show a fluorescent image of the samples at day 0 (as control). In general, how to distinguish the fluorescence by GFP-producing spheroids with the possible auto-fluorescence of cellulose?
5. Repair of BC materials: In addition to anisotropic growth, is the crystalline content different in BC spheroids and BC pellicles? One would also expect the pellicles to have a much higher concentration of BC compared to the spheroids, reducing migration of bacteria into the pellicles. These points should be discussed.
6. In the mechanical tests, do the repaired pellicles break at the repair sites? It would make sense to also measure the mechanical properties of films made from spheroids only, as another important control.
7. For storage of living spheroids, would it be possible to keep the bacteria alive for longer by freezing the spheroids?
8. Minor points: Several images are missing scale bars; in line 224: figure 2d does not exist.

Reviewer #2:

Remarks to the Author:

The manuscript titled "Bacterial cellulose spheroids as building blocks for 3D and patterned living materials and for regeneration" by Ellis et al. impressively described a reproducible protocol for culturing bacterial cellulose (BC) spheroids. In contrast to the anisotropic grown BC pellicles, the BC spheres' unique isotropic growing property allowed them to grow together, repair damaged cellulosic membrane, and act like glues to bind synthetic material fragments together. The authors exploited the obtained BC spheroids as modular building blocks to construct 2D/3D shapes and different patterns by manually arranging spheroids rather than using specific equipment-required molding and 3D printing techniques. A chemically-responsive barcode was fabricated via patterned BC spheroids as a material application. In my view, this is a nice piece of work that timely contributes to the emerging field of engineering living materials (ELMs). The reviewer has several issues that need to be addressed before the paper can be accepted for publication.

1. The reviewer suggest getting higher-resolution images for figure 3c. In addition, can the authors just engineer the sender cells to express GFP only so as to better distinguish the distributions of the two strains used in this article.

2. The authors listed three main determining factors (the initial optical density, the culture container, and the culture media) for BC spheroid formation. The reviewer is curious to see if all three conditions are required simultaneously. Alternatively, which factor will be mostly critical for the formation of the described BC spheroids?
3. The reviewer was quite confused about the experimental results in Figure 3A and Figure 4I. A pattern containing both normal BC spheroids and BC spheroids made by GFP-tagged *K. rhaeticus* was fabricated to demonstrate the possibility of creating ELMs functionalized at the millimeter scale (Figure 3A). The author explained that *K. rhaeticus* is a non-motile bacterium and will stay where it was. However, to explain that purified sterile BC spheroids successfully repaired punctures in BC pellicles, the author claimed that "bacteria from the fresh pellicle are migrating to the spheroids and producing the new cellulose." The reviewer thus was confused as there seems a contradiction between these two statements. Could the author clarify the confusions?
4. The author proved that BC spheroids could fuse sections of various materials (sponge, wood, and cotton) together. Can the authors describe the strengths of bonding in these cases?
5. In Line 227, the author quotes Figure 2D, but there is no D panel in Figure 2.
6. Three recently published review articles related to living materials are highly recommended to cite to help readers to understand the field of engineered biomaterials:
 - a) Engineered Living Materials: Taxonomies and Emerging Trends. Trends in biotechnology. 2020
 - b) Materials design by synthetic biology. Nature Reviews Materials. 2020
 - c) Stimuli-Responsive Engineered Living Materials. Soft matter. 2020

REVIEWER COMMENTS.

Reviewer #1 (Remarks to the Author):

The authors present a very interesting study on the preparation and use of bacterial cellulose (BC) spheroids for patterning and regeneration of engineered living materials. This is a highly interesting topic that has a lot of potential yet to be explored. The concept proposed is to use spheroids as building blocks to make living materials with genetically programmed patterns and 3D shapes. Exploratory experiments show that a low initial cell density during culture and small culture tubes are needed to produce BC spheroids by shaking. The authors show that BC spheroids can be programmed to send and receive chemical signals, encode information that can be retrieved upon chemical activation, and repair, regenerate and glue different materials. This study is expected to motivate a lot more work in this growing research field. Before acceptance of the manuscript, the following points need to be addressed:

We thank the reviewer for their work in reading and critically assessing our manuscript. We are glad that they see the potential of our approach and found the work highly interesting.

1. In the introduction, the authors state that additive manufacturing “requires washing the printed matrix to obtain the final shape”. Since no washing is needed in the method reported in Ref 19, please indicate another reference for previous work where this step is needed.

The reviewers are correct that no washing is required, and so we have corrected this now. In the work of reference 19, a FLINK matrix is used a 3D printable scaffold upon which to grow *A. xylinum* to produce cellulose in the shape of printing. FLINK is composed of biocompatible hyaluronic acid (HA), κ-carrageenan (κ-CA), and fumed silica (FS) but the ability of *Acetobacteriaceae* to degrade any of these compounds has not been reported. As such the final shape and material is only partially a product of the bacteria, with the majority of mass being the chemical scaffold. The final bacterially-made shape/material is never removed from within this scaffold by degradation or washing.

To better explain our point we have now modified the associated text for more clarity (line 62).

2. The use of low bacterial densities as a criterion to form spheroids is reasonable. But why does spheroid formation depend on the volume and shape of the culture tube ? Is the volume of the tube expected to change the oxygen concentration ? Or is this a purely mixing/hydrodynamic effect ? To allow researchers to reproduce these results, it is important that the authors provide more detailed information on the stirring conditions, e.g. shaker type, model, power, etc.

From the results of our experiments, media motion looks to be crucial to allow cellulose to grow in spheroid shapes. The oxygen concentration might be affected by motion as well, but we tried different shallow cultures in big flasks to increase oxygen concentration and had no success. At the same shaking speed, an increase in the flask volume implies an increase of diameter, affecting the forces transmitted to the liquid and its motion. As the reviewer states, the hydrodynamic effect produced by the container is likely to be critical to entangle the cellulose bands/fibers into spherical shapes.

We modified the text (line 92-102) to better convey our understanding to readers. We also provide more specific details about the equipment we have used in the Methods (line 398).

3. The barcode demonstration is very nice. However, for how long would one be able to keep this functionality considering that the bacteria eventually die upon storage?

The barcode functionality and some of the other applications we propose are dependent on the bacteria surviving. Their population within the material and thus their output signal will decay over time depending on the storage conditions. This can be seen in our results in Supplementary Figure 4 where we observe the bacteria dying over time in the material. The reviewer is correct in anticipating that this would be an inconvenient for certain ELMs applications. As we have been concerned about this, we have developed a solution to store these materials and maintain the bacteria alive for longer. For long term storage, we have tested a method to freeze bacteria to maintain cell viability, this is inspired by the general procedure used by most labs to conserve strain collections. We soak BC material in 15% glycerol solution for 1 h to allow glycerol to diffuse through the material. This can then be frozen at -80 C. We counted CFU before glycerol treatment and before-and-after freezing for up to 1 month of storage. We demonstrate that this method is effective for conserving bacteria alive in the material, with the viable cells remaining within an order of magnitude.

We have added this new experiment as Supplementary Figure 7, and in the revision we describe it the methods section (lines 477-481) and discuss the results in the main manuscript (lines 319-325, 358) and supplementary materials (Supplementary Figure 7).

4. Figure 4H: To ensure that the fluorescent signal comes from the GFP-expressing bacteria, wouldn't it be important to show a fluorescent image of the samples at day 0 (as control). In general, how to distinguish the fluorescence by GFP-producing spheroids with the possible auto-fluorescence of cellulose?

Apologies, but we must not have been clear in our explanation of these data. The bacteria producing these spheroids express GFP constitutively, therefore on day 0, spheroids are already expressing GFP. Cellulose itself shows some green autofluorescence, but the intensity of this signal is minimal in comparison to the high fluorescent signal of GFP produced by the cells. In image 4H, the threshold of the fluorescent signal intensity was set to remove the autofluorescence signal from the purified punctured cellulose. Therefore, the fluorescent signal in the picture is due only to intense signal from GFP. Figure 4B is a clear example to show that BC autofluorescence is very low compared to GFP made by the cells.

Text in the results sections (lines 158-159, 285-286) has been modified to better explain this.

5. Repair of BC materials: In addition to anisotropic growth, is the crystalline content different in BC spheroids and BC pellicles?

The crystallinity content of BC spheroids is an interesting question. To our knowledge, this hasn't been studied yet for spheroids. We are interested in doing this experiment and addressing this question. Unfortunately, after much discussion at our institute from March to June 2020, we were not allowed to access to the required X-Ray facilities due to current COVID restrictions which are not expected to ease until Autumn. This is a shame, however we do not think these data are required to support the claims of our of current work, which is focused on the macro-structure design using building blocks and their functionalization.

Our expectation is that crystallinity will not be different to what is seen in the pellicles. Cellulose crystallinity mainly depends on state of the fibres extruded from the cell by the synthesis complex formed by the proteins BcsAB,C,D in the cell membranes. We have no reason to believe that the changed culture conditions will alter the expression and assembly of this complex because in past published works the difference in BC crystallinity between static and shaking conditions was only 1-2% (Li J *et al.* Biotechnol Appl Biochem. 2019 Jan). The culture conditions used in that study are similar to those used in our work and we would expect similar variations in our approach.

One would also expect the pellicles to have a much higher concentration of BC compared to the spheroids, reducing migration of bacteria into the pellicles. These points should be discussed.

We could not see a reason ourselves for why pellicles and spheroids would have notably different BC concentrations as the reviewer expects. So to check this, we measured the BC concentration of pellicles and spheroids, obtaining values of 4.2% and 3.9% w/w of BC/water, respectively. We do not think that this difference is significant, and now have reason to believe other points explain the migration into hydrated sterile pellicles seen in the paper (as discussed in Lines 324-331). Ultimately, due to the very high water content of fresh BC, we expect that there is similar space in both BC types for the bacteria to enter and colonize the material.

6. In the mechanical tests, do the repaired pellicles break at the repair sites ? It would make sense to also measure the mechanical properties of films made from spheroids only, as another important control.

Our results show that repaired material is not as good as the original material and, as expected, samples always brake in the repaired area where the spheroids grew. This can be observed in the picture below (image **A**) showing three of the measured repair experiments. The brighter area corresponds to the spheroid-repaired region. The BC strips used for mechanical property assessment split in the middle of the repaired circle in a way that the center of the strip only contains cellulose from spheroids repair. A pellicle only made by spheroids will behave in the same way as the repaired material in our experiments due to the way we cut the samples. Our experiments are therefore actually just measuring the mechanical properties of a pellicle made of spheroids, as the original cellulose is working as a connector to hold a small region made of spheroids (as show in the scheme in image **B**). This is now clarified on Line 248 of the manuscript

7. For storage of living spheroids, would it be possible to keep the bacteria alive for longer by freezing the spheroids ?

We appreciate this comment, and we considered it a great idea to explore. As mentioned in the reply to comment 3, we have tested a method for BC ELMs freezing inspired by microbial culture storage. We anticipated that simple freezing of the pellicles would compromise cell survival and ice crystal growth would kill cells and may even damage BC. The use of glycerol in the media solves this problem.

We have added the information related to this experiment to the methods section (lines 477-481) and supplementary materials (Supplementary Figure 7). We discuss the results in the main manuscript (lines 319-325, 358).

8. Minor points: Several images are missing scale bars;

We have included a scale bar in every image.

in line 224: figure 2d does not exist.

Apologies, it should be 3B. It has now been corrected.

Reviewer #2 (Remarks to the Author):

The manuscript titled "Bacterial cellulose spheroids as building blocks for 3D and patterned living materials and for regeneration" by Ellis et al. impressively described a reproducible protocol for culturing bacterial cellulose (BC) spheroids. In contrast to the anisotropic grown BC pellicles, the BC spheres' unique isotropic growing property allowed them to grow together, repair damaged cellulosic membrane, and act like glues to bind synthetic material fragments together. The authors exploited the obtained BC spheroids as modular building blocks to construct 2D/3D shapes and different patterns by manually arranging spheroids rather than using specific equipment-required molding and 3D printing techniques. A chemically-responsive barcode was fabricated via patterned BC spheroids as a material application. In my view, this is a nice piece of work that timely contributes to the emerging field of engineering living materials (ELMs).

We thank the reviewer for their work in reading and critically assessing our manuscript. We are glad that they see this as a timely contribution the field.

1. The reviewer suggest getting higher-resolution images for figure 3c.

We agree with the reviewer that some of the pictures were not totally clear. We have now changed the 'Day 0' pictures for other images taken at the time from the same experiment which have better focus. We believe this substantially improves the clarity.

In addition, can the authors just engineer the sender cells to express GFP only so as to better distinguish the distributions of the two strains used in this article.

We thank the reviewer for this comment. Indeed, further engineering on the bacteria to express GFP alongside cells expressing RFP would be desirable to produce a clear two-colour pattern. However, we elected not to do this as a sender strain made with a GFP-expression cassette also added into the same plasmid had previously shown reduced growth rate and material production due to the burden of GFP expression. This strain can be potentially engineered to express GFP at a lower level (e.g. by tuning the RBS strength) to reduce the burden on growth, but this would be at the cost of the output fluorescence being close to the autofluorescence of the cellulose. Ideally we would explore using a different coloured fluorescent protein, such as BFP for these patterns, but as it stands we only have 2 options available: RFP and GFP.

Because of this we elected instead to make an experiment with just RFP-expressing and non-fluorescent cells and instead show patterning by showing different spheroids configurations. In the figure, we show three different pictures. In the single spheroid sender-receiver, we show how single spheroids activate each other. In the circular distribution of spheroids, we show how the signal diffuses to other spheroids, even to those spheroids in the central area that are not in contact with the surrounding sender spheroids. In the third picture, the square distribution allows distinguishing the distribution of the sender and receiver as a circular distribution could be confusing.

We have modified the text (Lines 176-179) to highlight the information that can be obtained from each spheroid configuration. In the future our new toolkit for engineering BC-producing strains (<https://www.biorxiv.org/content/10.1101/2021.06.09.447691v1> will hopefully enable pigmented patterns)

2. The authors listed three main determining factors (the initial optical density, the culture container, and the culture media) for BC spheroid formation. The reviewer is curious to see if all three conditions are required simultaneously. Alternatively, which factor will be mostly critical for the formation of the described BC spheroids?

The initial optical density and the culture container are critical conditions that are required simultaneously based on preliminary work we did at the start of the project. As described in the manuscript, media composition affected the number and size of spheroids, but was not a critical determinant factor for BC spheroids production.

We have modified the text (lines 92-105) to clarify this point.

*3. The reviewer was quite confused about the experimental results in Figure 3A and Figure 4I. A pattern containing both normal BC spheroids and BC spheroids made by GFP-tagged *K. rhaeticus* was fabricated to demonstrate the possibility of creating ELMs functionalized at the millimeter scale (Figure 3A). The author explained that *K. rhaeticus* is a non-motile bacterium and will stay where it was. However, to explain that purified sterile BC spheroids successfully repaired punctures in BC pellicles, the author claimed that "bacteria from the fresh pellicle are migrating to the spheroids and producing the new cellulose." The reviewer thus was confused as there seems a contradiction between these two statements. Could the author clarify the confusions?*

Many bacteria possess different forms of motility like swimming, twitching or swarming that allow them to actively move even dozens of centimeters overnight in a Petri dish culture. BC-producing

bacteria like *K. rhaeticus* do not appear to have any specific motility apparatus (e.g. flagella) and so are unable to achieve movements on that scale. However, as we have seen in our time lapse experiments, this bacterium can move slowly at the microscale as it synthesizes cellulose, using the cellulose production to propel itself. This effect was studied extensively by Malcolm Brown (UT Austin - <https://doi.org/10.1073/pnas.73.12.4565>). This movement is likely to be enough to migrate to local spaces and colonize them, for example when bacteria are placed into a gap within a purified BC piece. The microscale movement is enough for cells to move into the BC nanostructure and aid repair, but macroscopically a seeded pattern on the cm scale will remain intact.

In Figure 4I, we show how migration of cells can go from pellicle to spheroids. On reflection, we now think that the reviewer is right that migration might not be the only explanation for repair with purified spheroids. Instead it may be the case that as sterile purified spheroids are added to the punctured pellicle, they get coated with live bacteria in the culture media who then established themselves on the spheroid and continue its growth and connection with the pellicle. To demonstrate this, we have added a final experiment, where a punctured sterile pellicle is repaired with sterile spheroids when live bacterial cells are added from the liquid culture.

We have changed the manuscript text (lines 153, 312-317) to include these new data, cite the key work from Brown and discuss the findings. New data have been added to the supplementary material (Supplementary Fig 6D and Supplementary Video 4).

4. The author proved that BC spheroids could fuse sections of various materials (sponge, wood, and cotton) together. Can the authors describe the strengths of bonding in these cases?

The tensile strength of the fused materials (synthetic blue sponge, cotton and Luffa sponge) with BC spheroids was measured, obtaining different values depending on the material (0.1-5MPa) and revealing low tensile strength values compared with the outstanding properties of BC material. The fused material always broke at the joint, indicating that the integration method could be improved, for instance, in the edge shape to increase surface contact. We have commented these results in the manuscript and added the new data to supplementary material (**Supplementary Figure 5B**).

5. In Line 227, the author quotes Figure 2D, but there is no D panel in Figure 2.

Apologies, this should say 3B. It has been corrected.

6. Three recently published review articles related to living materials are highly recommended to cite to help readers to understand the field of engineered biomaterials:

- a) Engineered Living Materials: Taxonomies and Emerging Trends. Trends in biotechnology. 2020*
- b) Materials design by synthetic biology. Nature Reviews Materials. 2020*
- c) Stimuli-Responsive Engineered Living Materials. Soft matter. 2020*

Thank you for the suggestions, we have now included these citations into the paper as citations 2, 4 and 33.

Reviewers' Comments:

Reviewer #1:

Remarks to the Author:

The authors have carefully addressed all the remarks from the reviewers. I recommend acceptance of the manuscript as is.

Reviewer #2:

Remarks to the Author:

The authors have thoroughly addressed the questions and concerns raised by the reviewers. The reviewer here recommends that the manuscript can be accepted for publication.